

Changes in tropical cyclones under stabilized 1.5ºC and 2.0ºC global warming
scenarios as simulated by the Community Atmospheric Model under the HAPPI
protocols
Michael F. Wehner[1*], Kevin A. Reed[2], Burlen Loring[1], Dáithí Stone[1], Harinarayan
Krishnan[1]
[1] Lawrence Berkeley National Laboratory, Berkeley, California 94720, USA
[2] State University of New York at Stony Brook, Stony Brook, New York 11794, USA
* Corresponding Author: mfwehner@lbl.gov

## Abstract

The United Nations Framework Convention on Climate Change (UNFCCC) invited
the scientific community to explore the impacts of a world where anthropogenic
global warming is stabilized at only 1.5ºC above preindustrial average temperatures.
We present a projection of future tropical cyclone statistics for both 1.5ºC and 2.0ºC
stabilized warming scenarios by direct numerical simulation using a high resolution
global climate model. As in similar projections at higher warming levels, we find that
even at these low warming levels the most intense tropical cyclones becomes more
frequent and more intense, while simultaneously the frequency of weaker tropical
storms is decreased. We also conclude that in the 1.5ºC stabilization, the effect of
aerosol forcing changes complicates the interpretation of greenhouse gas forcing
changes.
## Introduction
Changes in tropical cyclone intensity, frequency and distribution are expected as the
climate warms due to anthropogenic changes in the composition of the atmosphere.
While the development of a complete climate theory of tropical cyclones remains
elusive (Walsh, et al. 2015), recent advances in high performance computing
enables multi-decadal simulations of climate models at tropical cyclone permitting
resolutions. Together with conceptual models, such numerical models are the tool of
choice for investigating projected future changes in tropical cyclones (Wehner et al.
2017a).
Previous work has studied the impact of climate change on tropical storms through
idealized representations of future climate through uniform increases in greenhouse
gases and sea surface temperature (Walsh et al. 2015, Wehner et al. 2015) or more
realistic but more extreme cases of warming using the Representative
Concentration Pathways (RCP4.5 or RCP8.5) scenarios (e.g., Camargo, 2014;
Knutson et al. 2015; Bacmeister et al., 2016). The United Nations Framework
Convention on Climate Change (UNFCCC) invited the International Panel on Climate
Change (IPCC) to explore the impacts of a world where the expected average
warming remains less than or equal to 2.0ºC over preindustrial levels. In particular,
the UNFCCC requested an analysis of the feasibility and impacts of a target stabilized
global mean temperature of 1.5ºC over preindustrial levels. The Half A degree
additional warming, Prognosis and Projected Impacts (HAPPI) experimental



protocol was designed in response to this request to permit comparison of the
effects of stabilizing anthropogenic global warming at 1.5$^o$C over preindustrial levels
to 2.0$^o$C (Mitchell et al. 2017). In this paper, we present results from a high
resolution atmosphere-land model forced by the HAPPI prescriptions of sea surface
temperature (SST) and sea ice concentration.
The HAPPI experimental protocol consists of three parts (Mitchell et al. 2017). The
"Historical" part specifies observed sea surface temperatures (SST) from the NOAA
OI.v2 gridded monthly mean observational product (Reynolds et al. 2002) over the
period 1996-2015. An estimate of SST and sea ice concentrations in stabilized
scenarios at both 1.5$^o$ and 2.0$^o$C are constructed from the CMIP5 (Coupled Model
Intercomparison Project) multi-model database of future climate projections under
the RCP2.6 and RCP4.5 forcing scenarios hereafter designated "HAPPI15" and
"HAPPI20". These surface forcing functions are constructed using the observations
from 2006-2015 to preserve observed interannual variations. As such, Historical
year 2006 is directly comparable to HAPPI15 or HAPPI20 year 2106 as the date in
the stabilized scenarios is arbitrarily increased by a century. The original design of
the HAPPI protocols follows that of the "Climate of the 20$^{th}$ Century Plus Detection
and Attribution project" (C20C+) (Stone et al. 2017a) and targets large ensembles of
50 realizations or more to quantify the differences in projections (or attribution) of
extreme events in specific years. However, at the high horizontal resolutions
necessary to simulate tropical cyclones, the computational costs of the climate
model are too high to permit such a large number of simulations and ensemble sizes
are restricted. Hence, in this study we pool results across both simulation years and
the ensembles for each part of the HAPPI experiment to isolate the climate change
signal, if any, from internal variability. As part of our participation in the C20C+
project, we began the Historical simulation period in 1996 extending through 2015
thus permitting a more robust estimate of present day simulated tropical cyclone
statistics for comparison to the stabilized warmer climate.
This study uses the Community Atmospheric Model version 5.3 configured at a
global resolution of approximately 0.25° roughly equaling a grid spacing of 28 km in
tropical regions.  Note that this participating model is listed as "CAM5.1.2-
0.25degree" in the HAPPI documentation (http://portal.nersc.gov/c20c/data.html),
but here is abbreviated to "CAM5". This configuration has been demonstrated to
produce reasonable annual numbers of tropical cyclones at the global scale
compared to observations (Bacmeister et al. 2014; Zarzycki et al. 2014; Wehner et
al. 2014; Reed et al. 2015). The formulation of the dynamical core portion of the
atmospheric model does influence tropical cyclone counts and intensities (Reed et
al. 2015). The model used in this study used CAM5's finite volume based dynamical
core on a latitude-longitude grid (Lin and Rood 1996; Lin and Rood 1997;Lin 2004).
Storms up to category 5 on the Saffir-Simpson scale are regularly produced allowing
investigation into the effects of global warming on the distribution of tropical
cyclone intensity. The relationship between maximum wind speed and central
pressure minima was also demonstrated to be realistic (Wehner et al. 2014).
However, there are significant biases in track and cyclogenesis density, particularly



in the Pacific Ocean with the model simulating too many storms in the central North
Pacific and too few in the northwestern part of that basin.
Nonetheless, the high-resolution CAM5 can be a informative tool to explore the
change in tropical cyclone behavior in altered climates. Wehner et al. (2015)
explored tropical cyclone behavior in the four idealized climate change
configurations of the US CLIVAR Hurricane Working Group (Walsh et al. 2015). That
project compared the combined effect of a spatially uniform 2°C increase applied to
a climatological average of observed SST centered at 1990 and of a doubling of
atmospheric $CO_2$ to a control 1990 simulation, as well as the separate effects of each
factor. Their principal finding was that a lower resolution (1°) version of the CAM5
as well as methods based on the Genesis Potential Index (Emanuel and Nolan 2004)
could not reproduce the sign of the change in the global number of tropical cyclones
produced by the high resolution version. Under the combined effect of the uniform
2°C SST increase and $CO_2$ doubling, the high resolution CAM5 reduced the annual
number of tropical storms (categories 0–5) from 86±4 to 70±3. However, the annual
number of intense tropical cyclones (categories 4–5) increased from 10±1.7 to
12±1.7. The two separate forcing simulations revealed that most of the reduction in
the total number of tropical storms of all intensities was caused by the change in the
vertical temperature profile due to the $CO_2$ doubling while the increase in the
number of intense tropical cyclones was caused solely by the increased SST. The
warmer SST conditions also caused the maximum wind speeds of the most intense
storms to increase and their central pressure minima to decrease while $CO_2$
doubling had the opposite effect. The peak of the zonally averaged tropical storm
track density shifted poleward by ~2° in the Northern Hemisphere and ~4° in the
Southern Hemisphere in all three perturbed US CLIVAR configurations. A small
poleward shift (~1°) in Northern Hemisphere cyclogenesis origins was exhibited in
the two simulations with warmer SSTs but not the $CO_2$ doubling only simulation,
while all three perturbed simulations exhibited a similar shift in the broader
Southern Hemisphere cyclogenesis distribution.
The SST and sea ice perturbations imposed by the HAPPI protocols exhibit the more
realistic spatially varying SST patterns shown in Figure 1 than the uniform increase
of the US CLIVAR experiments. In the HAPPI protocols, warmer configurations are
produced by adding monthly climatological perturbations to the observed SSTs for
each individual month, preserving the current patterns of SST variability. The SST
perturbations for the 1.5°C stabilization scenario are taken directly from the
multimodel mean of CMIP5 RCP2.6 simulations (which conveniently warm by
approximately that amount on average above pre-industrial temperatures).
Radiative forcings (greenhouse gas concentrations, burdens of various aerosol
species, and ozone concentrations) are also taken directly from the RCP2.6 values.
The 2.0°C scenario uses SST perturbations and $CO_2$ concentrations interpolated
between CMIP5 RCP2.6 and RCP4.5 multi-model means, while other forcings remain
the same as for the 1.5°C scenario. Sea ice concentrations are computed using an
adapted version of the method described in Massey (2017) by using observations of
SST and ice to establish a linear relationship between the two fields for the time





period 1996-2015 and are consistent with the HAPPI prescribed SST fields. Details
are further described in Mitchell et al. (2017). Although they represent a smaller
perturbation to the climate system than does the US CLIVAR experiment, the HAPPI
experiment is more physically consistent in terms of the relationship of the SST
change to radiative forcing changes and in the distribution of sea ice in the high
latitudes permitting the HAPPI simulations to be more widely applicable to
phenomena outside of the tropics.
The CAM5 simulations performed for the HAPPI project consist of 5 realizations of
the Historical period plus 6 realizations of each stabilization scenario. One of the
Historical realizations is incomplete due to computer resource limitations resulting
in 96 simulated years for this part of the dataset. Sixty simulated years were
produced for both the 1.5 and 2.0$^o$C stabilization scenarios. Data products are freely
available with further information provided at www.portal.nersc.gov/c20c.
Simulated tropical cyclones are identified and tracked with the Toolkit for Extreme
Climate Analysis (TECA2.) available for download and installation at
https://github.com/LBL-EESA/TECA using the methods described in Knutson et al.
18  (2007).



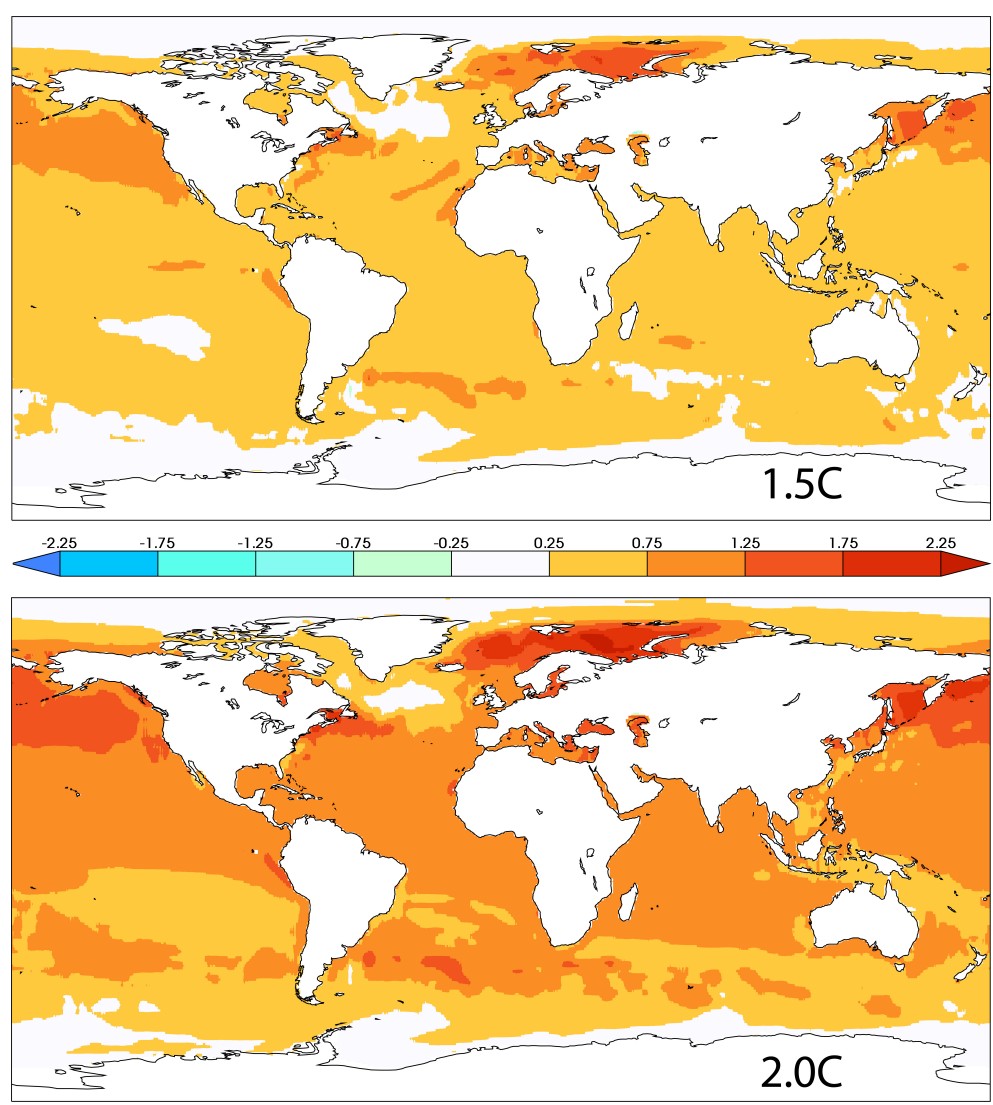

Figure 1: The temporal average of the imposed change (°C) in sea surface
temperature as prescribed by the HAPPI protocols. Upper. 1.5°C stabilization.
Lower: 2.0°C stabilization.
Another critical difference between the HAPPI and the US CLIVAR experimental
protocols is the aerosol forcing. While the US CLIVAR protocols had no specified
changes to aerosols, the HAPPI protocols set aerosol forcings to the end of the 21st
century levels under the RCP2.6 scenario for both stabilization scenarios. Hence,
there is a substantial reduction in the aerosol forcing in the stabilization simulations
compared to the Historical simulations. Dunstone et al. (2013) indirectly found a
substantial reduction in Atlantic tropical storms by varying aerosol forcing in the UK



MetOffice climate model HadGEM2-ES at a resolution of 1.2°x1.9°. In the CAM5
simulations presented here, we used its bulk aerosol model to prescribe aerosol
concentrations rather than emissions in order to reduce the computational burden
(Kiehl et al., 2000). Huff et al. (2017) established that CAM5 does exhibit sensitivity
to aerosol formulation in the simulated number and intensity distribution of tropical
cyclones in the simulated current climate. However, the HAPPI protocol does not
establish a controlled investigation of the effects of the aerosol forcing reduction in
the stabilized scenarios nor have we performed such simulations yet. Figure 2
shows the percent change in total aerosol optical depth in the visible band
comparing the Historical and 2.0°C stabilization simulations averaged over all years
and realizations. Significant decreases are evident over most of the entire Northern
Hemisphere and tropics. Results from the 1.5°C stabilization simulations are the
same.

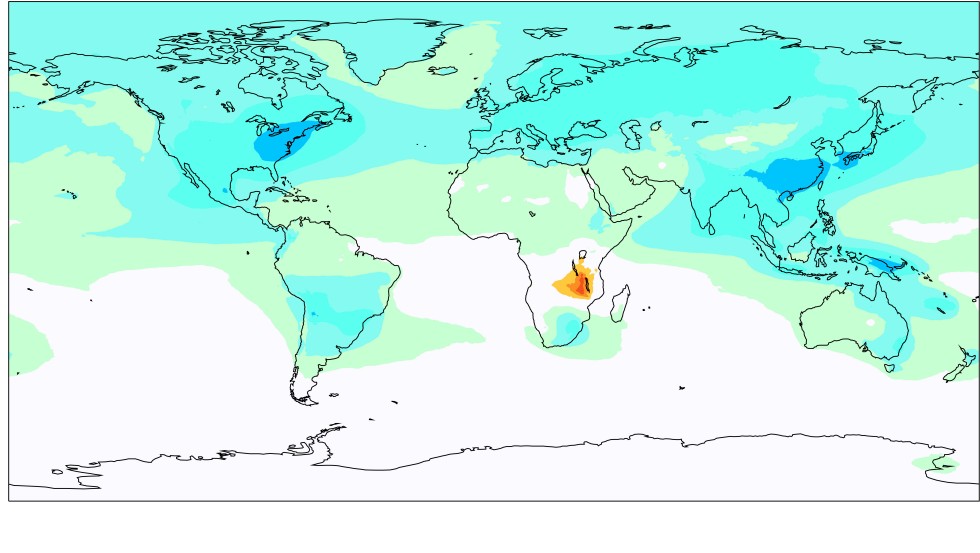

Figure 2: Percent difference between the stabilized 2°C scenario and the historical
simulation of the total aerosol optical depth in the visible band.
**Results**
As in the US CLIVAR idealized experiments, the global number of intense tropical
cyclones (category 4 and 5) is substantially increased in the warmer climates of the
HAPPI stabilization scenarios with a statistical significance higher than the 1% level
as shown in Figure 3. Also as in the idealized warming experiments, the number of
tropical storms (category 0) is substantially decreased in a warmer climate.
However, the effect on the total number of named storms of all intensities (category
0-5) is subtler in the HAPPI simulations. For this version of CAM5, the global annual





number of category 0 to 5 storms is 73.4±0.91 in the Historical ensemble[1]. In the
1.5°C stabilization scenario, this number is only reduced to 72.5±1.2, which is not
significant at a 10% significance level. However, in the 2.0°C stabilization scenario, a
further reduction to 67.5±1.3 is realized which is significant at the 1% level. In the
cooler stabilization scenario, the decrease in category 0 storms is roughly offset by
the increase in intense storms leading to the insignificance of the change in the total
number of storms. In the warmer scenario, the yet larger decrease in category 0
causes the change in the total number of storms to be more significant. In both
stabilization scenarios, the changes from the Historical simulation in categories 1,2
and 3 storms are not statistically significant above the 5% level. Differences
between the 1.5°C and 2°C stabilization scenarios are only highly significant in the
decrease by category for the number of the weakest category of storms.
Importantly, the differences in the number of intense tropical cyclones between the
two warming scenarios are not statistically significant in this study. The results
presented in Figure 3 are repeated numerically in table 1.

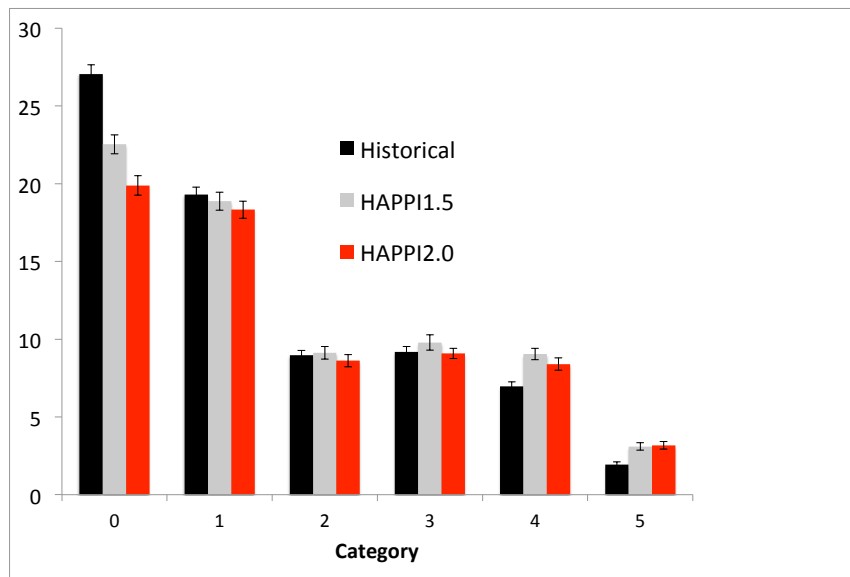

Figure 3. Global annual number of tropical cyclones by Saffir-Simpson scales for the
historical (black), 1.5°C stabilization scenario (gray), 2°C stabilization scenario

---

[1] The Historical annual global tropical storm counts over all categories differs from
the 1990 climatological simulations of Wehner et al. (2015) for three reasons. 1)
SST are a slightly different 2) The version of CAM5 is a more recent release (CESM
v1.2.2 vs. v1.0.3) 3) Subtle differences in the implementation of the tracking
algorithm.





(red). Error bars are the standard errors. Black: Historical. Gray: 1.5º Stabilization.
Red: 2.0º Stabilization.

| Saffir-Simpson | 0-5 | 0 | 1 | 2 | 3 | 4 | 5 |
|---|---|---|---|---|---|---|---|
| HAPPI15 minus Hist | **-0.9** | -4.5 | -0.4 | 0.2 | 0.6 | **2.1** | **1.2** |
| HAPPI20 minus Hist | **-5.9** | **-7.2** | *-1.0* | -0.4 | -0.1 | **1.4** | **1.2** |
| HAPPI20 minus HAPPI15 | **-5.0** | **-2.6** | -0.5 | -0.5 | -0.7 | -0.6 | 0.1 |

Table 1. Differences in CAM5 simulated global annual tropical storm counts by
Saffir-Simpson scale between the two HAPPI stabilization scenarios and the
Historical simulation and each other. Differences that are statistically significant at
the 1% level are in bold while those at the 10% level are in italics.
Average storm track length, duration and mean translational speed are shown for
the HAPPI scenarios as a function of maximum lifetime intensity on the Saffir-
Simpson scale in Figure 4. Weak storms (category 0) show no substantial changes in
track length, translational speed or duration between the three ensembles of CAM5
simulations and this result is consistent with the US CLIVAR experiments (Wehner
et al. 2015). While these three metrics show increases for Category 2-4 storms in the
1.5ºC stabilization scenario compared to the Historical simulations, those increases
are attenuated in the warmer 2.0ºC stabilization scenario. However, the most
intense storms (category 5) exhibit consistent increases in track length and duration
on average as the climate system warms. Translational speed (here averaged over
the entire storm duration) increases in all three ensembles with storm intensity but
the differences between scenarios is complex. Notably, while increases in average
translational speed in the warmer scenarios are simulated for storms in the middle
of the Saffir-Simpson scale, decreases are simulated for the most intense category.
While all of the differences in Figure 4 are statistically significant well above the 1%
level due to the large number of storms tracked, subtle changes in the experimental
design, including changes in SST pattern or aerosol forcing might alter these results.
Better quantification of this type of structural uncertainty will require further
developments in high performance computing technologies to permit more diverse
experiments.

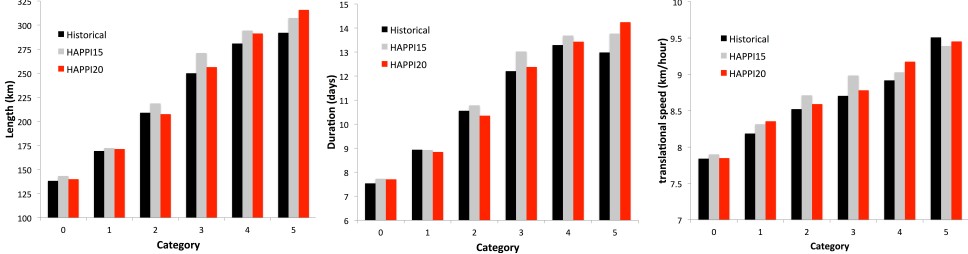

Figure 4: Left: Average tropical storm track length (km) for the HAPPI scenarios as a
function of maximum intensity on the Saffir-Simpson scale. Middle: Average tropical
storm track duration (days) for the HAPPI scenarios as a function of maximum
intensity on the Saffir-Simpson scale. Right: Average tropical storm track speed



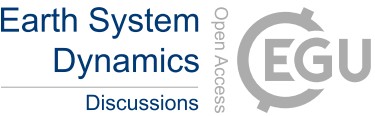

(km/hour) for the HAPPI scenarios as a function of maximum intensity on the Saffir-
Simpson scale. Black: Historical. Gray: 1.5° Stabilization. Red: 2.0° Stabilization.
The zonal average of the normalized density of storm tracks of all intensities for the
HAPPI scenarios is shown in the left panel of Figure 5. As mentioned above, CAM5 is
known to have a significant bias in the genesis location of Pacific tropical storms
although the total number, both in that basin and globally is not far from observed
records. More detailed but somewhat noisy maps of track density differences
between the HAPPI scenarios are shown in the Appendix in Figure A1. Integrating
over all longitudes, as in Figure 5, damps this noise revealing a poleward shift in the
warmer HAPPI scenarios compared to the Historical simulations. In the Northern
Hemisphere, there is a tendency for a substantially larger normalized density of
storm tracks poleward of 25N in both the Atlantic and Pacific Ocean basins (see
Figure A1). This may partially explain the increased track lengths and durations
shown in Figure 4. With warmer temperatures, conditions that can sustain tropical
storm wind speeds extend poleward. Although not considered here, there is
potential for an anthropogenic influence on the transition to extra-tropical
characteristics of storms that undergo them. In the Southern Hemisphere, Figure 5
reveals that normalized storm track density is a narrower function of latitude in the
warmer HAPPI scenarios. Figure A1 reveals that this is mainly due to a change in the
location of simulated tropical storms in the Southern Indian Ocean. In both
hemispheres, differences between the 1.5°C and 2.0°C stabilization scenarios is
smaller and noisier making any differences in track density difficult to interpret.
The zonal average of the normalized cyclogenesis density for tropical storms of all
intensities is shown in the right panel of Figure 5. Again, more detailed but noisy
maps of cyclogenesis density differences between the HAPPI scenarios are shown in
the Appendix in Figure A2. In the Northern Hemisphere, a much smaller poleward
shift than for track density starting at about 15N is simulated in the warmer HAPPI
scenarios compared to the Historical simulations. Figure A2 suggests that much of
this change is coming from the Atlantic Ocean but these cyclogenesis differences are
not as compelling as they are for the tropical storm tracks. In the Southern
Hemisphere, the cyclogenesis changes are similar to the track changes in both
Figure 5 and the Appendix. Hence, we can conclude that the shifts in Southern
Hemisphere tracks are mainly a result of cyclogenesis shifts that are mostly in the
Southern Indian Ocean.



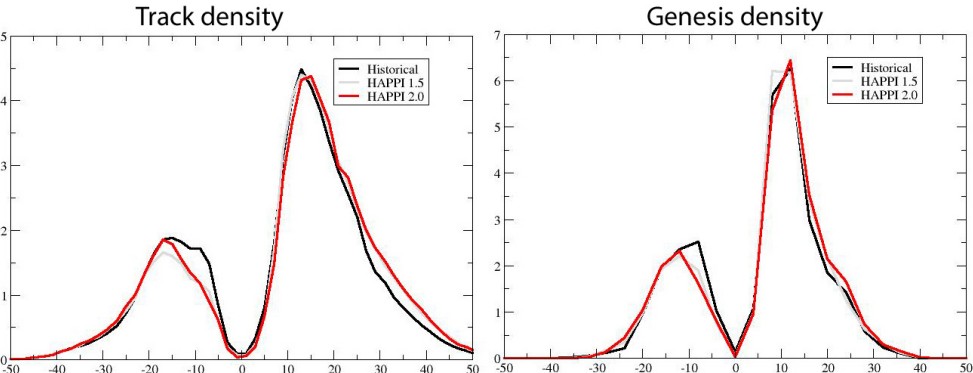

Figure 5. Left: Zonally averaged normalized tropical storm track density for the
HAPPI scenarios. Right: Zonally averaged normalized tropical storm genesis density
for the HAPPI scenarios. Black: Historical. Gray: 1.5º Stabilization. Red: 2.0º
Stabilization.
The annual Accumulated Cyclonic Energy (ACE) is shown in Figure 6 for the
Historical and HAPPI stabilization scenarios both globally and by the major ocean
basins with tropical cyclone activity. ACE is a measure of the annual kinetic energy
contained in tropical storms and is obtained by squaring the maximum sustained
surface wind in the system every six hours and summing it up for the year
(http://www.cpc.ncep.noaa.gov/products/outlooks/background_information.shtml
). Globally, ACE is mainly increased in the 1.5ºC stabilization scenario by the
increase in the number of intense tropical cyclones. Increases in average storm
duration also lead to in the increase in ACE. However, as the total number of storms
is significantly decreased in the 2.0ºC stabilization scenario, ACE is decreased
compared to the cooler stabilization scenario. The global changes are dominated by
similar changes in the North Atlantic and Northeast Pacific. Changes in the
Northwest Pacific do not exhibit large changes but CAM5 has a significant
cyclogenesis location bias in the Pacific Ocean that may be relevant. While the total
number of simulated north Pacific storms is a reasonable representation of
observations (Wehner et al. 2014), Northwestern Pacific storms originate too far to
the east causing cyclogenesis and track densities to be too high in the central Pacific
and is focus of current research to be presented elsewhere. Also of note is that ACE
in the Southern Indian Ocean does not change despite the cyclogenesis and track
changes discussed above.



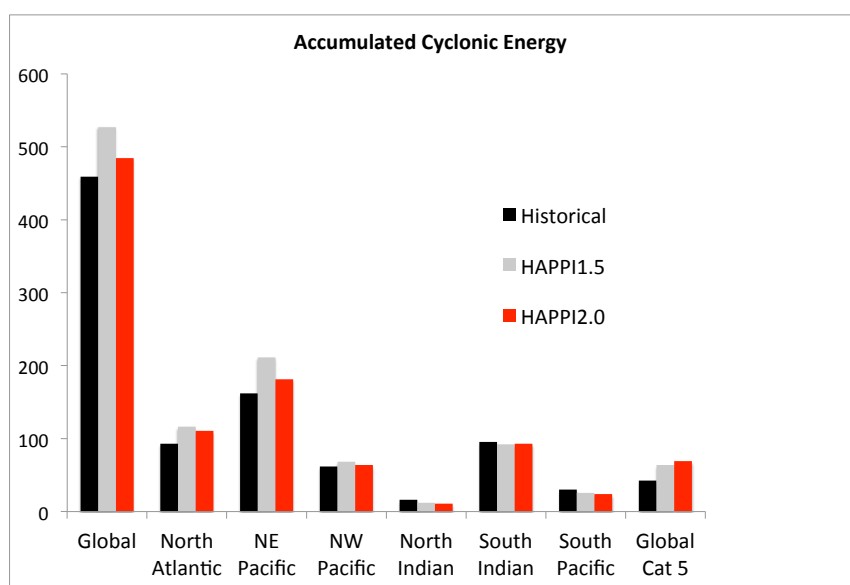

Figure 6. Average annual Accumulated Cyclonic Energy (ACE) for the Historical and
HAPPI stabilization scenarios for all named storms by basin and globally for intense
tropical cyclones only. Units: 1 ACE=$10^4$ knots. Black: Historical. Gray: 1.5°
Stabilization. Red: 2.0° Stabilization.
Figure 7 shows the relationship between peak wind speeds and central pressure
minima at the time of maximum intensity for the three HAPPI ensembles. As there
are no changes to the model configuration between the simulations other than
forcing conditions, this relationship does not change other than the appearance of
combinations of wind speed and pressure at the very highest simulated intensities
in the warmer simulations that do not occur in the Historical simulation. We
conclude then that warmer temperatures do not influence the relationship between
tropical storm peak wind speed and minimum central pressure. We do note
however that model structural changes can influence the simulation of this
relationship (Reed et al. 2015) thus requiring that evaluation of the effect of forcing
changes on tropical storm statistics must be only done with simulations from the
same version of the climate model.





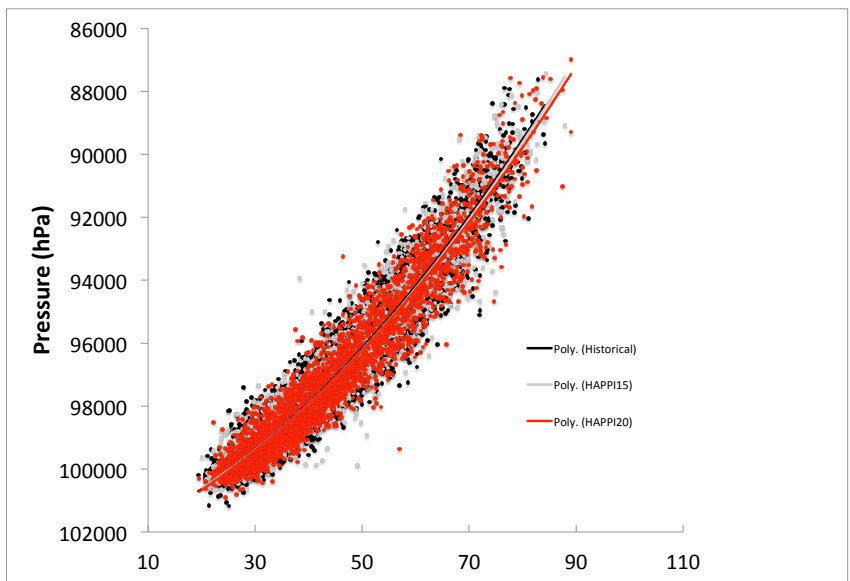

Figure 7: Scatterplot of minimum central pressure (hPa) versus maximum wind
speed (m/s) at the time of maximum intensity for the HAPPI simulations. Black:
Historical. Gray: 1.5º Stabilization. Red: 2.0º Stabilization. Solid lines are quadratic
fits to the data.
A definition of the physical size of tropical storms has recently been developed by
Chavas et al. (2016) by defining an approximate radius at specified wind speeds.
Figure 8 shows average Chavas radii for the Historical and HAPPI stabilization
scenarios. Radii are calculated every three hours over the duration of every tracked
storm for the threshold wind speeds defining the Saffir-Simpson categories as well
as for the storms' maximum wind speed. Each relevant radius is calculated for a
given storm. For instance, we calculate 6 radii for a category 5 storm (1 for each
Saffir-Simpson threshold) but only a single radius for a category 0 storm.  The CAM5
HAPPI simulations exhibit about a 5% increase in category 0 storm size and a
smaller (2-3%) increase in category 1 storm size in the warmer stabilized climates.
Little change in storm size is simulated for more intense tropical cyclones except for
category 5 storms in the 2ºC stabilization scenario that experience an 8% increase in
Chavas radius. The increase in weak storm size may be due to the change in the
track density discussed above. The increased fraction of tracked tropical storms at
higher latitudes are likely to be in the lower categories and may be starting their
extra-tropical transition but maintaining high winds. The increase in category 5
storm size in only the warmer of the two HAPPI stabilizations currently lacks an
explanation. Planned simulations of this version of CAM5 with the so-called
unHAPPI protocols (stabilized at 3º and 4ºC above preindustrial levels) may provide
some insight into these aspects of change in storm structure.




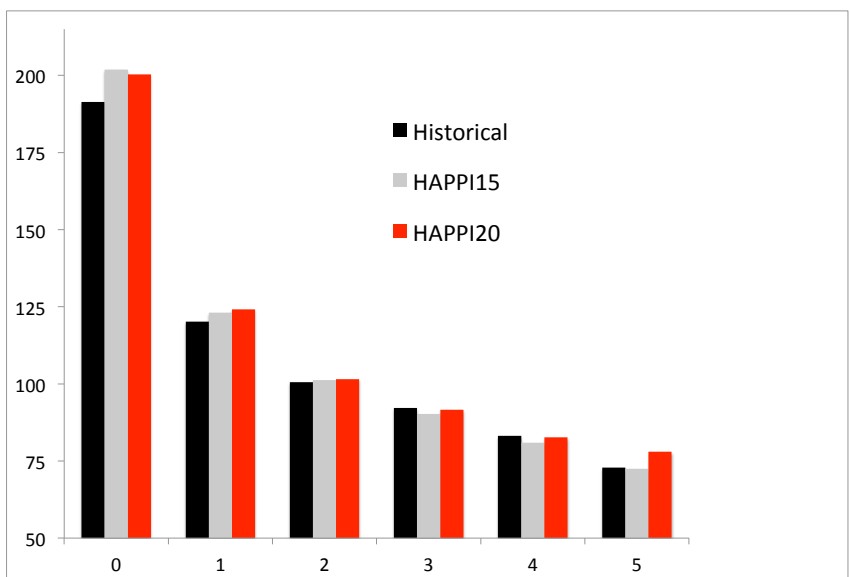

Figure 8. Chavas radii at different wind speeds selected as the definitions of the
Saffir-Simpson categories (km) for the HAPPI simulations. Black: Historical. Gray:
1.5° Stabilization. Red: 2.0° Stabilization.
**Conclusion**
The Half A degree additional warming, Prognosis and Projected Impacts (HAPPI)
experimental protocol was designed to rapidly inform the Intergovernmental Panel
on Climate Change about the differences between stabilized climate at 1.5°C and
2.0°C above preindustrial global temperatures. However, it does not isolate all of the
effects of forcing changes required to stabilize the climate from the present day
conditions. In particular, the effect of sulfate aerosol reductions in the atmosphere
has a non-local effect in the HAPPI simulations and has been demonstrated to be
important to assessing changes in tropical cyclones (Huff et al. 2017) and heat
waves (Wehner et al. 2017b). As the radiative forcing changes due to $CO_2$ between
the historical and 1.5°C scenarios may be smaller than the forcing changes due to
aerosols, the $CO_2$ effects in tropical storms may be comparable or even smaller by
the aerosol effects at this stabilization level.
It is fair to say that the simulated differences tropical cyclone statistics between the
1.5°C and 2.0°C stabilization scenarios as defined by the HAPPI protocols are small.
Indeed, both warmer climates produce fewer tropical storms over all intensities in
the global sense and the reduction increases as the sea surface temperature (SST)
becomes warmer. Also, the most intense storms become more intense in both
warmer SST configurations with the highest peak wind speeds and lowest central
pressure minima simulated in the warmer of two stabilizations.



Given the similarities between the two HAPPI scenarios and the importance of
aerosol forcings, a more complete understanding of tropical storm frequency in
aggressively stabilized climates requires detailed descriptions of the changes in
those forcings. This would be particularly critical in geoengineering schemes relying
on solar radiation management. However, as found by Bacmeister et al. (2016) in
their comparison of RCP4.5 to RCP8.5, major uncertainties in the pattern of SST
changes also pose a significant challenge in accurately projecting future tropical
storm frequency.
Changes in other important characteristics of tropical cyclone behavior are subtler.
Both warmer climate conditions considered here project significant changes in the
poleward density of tropical storm tracks compared to the Historical simulations
but the differences between them is not likely to be highly significant. Also, changes
in Accumulated Cyclonic Energy (ACE), storm duration, track length and
translational speed are complex with the differences clearly evident for only the
most intense storms. Finally, some properties of tropical cyclones are not
significantly altered in warmer climates, most notably the robust relationship
between maximum wind speeds and central pressure minima.

**Acknowledgement**
The work at LBNL was supported by the Department of Energy's Office of Science
under contract number DE-AC02-05CH11231. This document was prepared as an
account of work sponsored by the United States Government. While this document
is believed to contain correct information, neither the United States Government nor
any agency thereof, nor the Regents of the University of California, nor any of their
employees, makes any warranty, express or implied, or assumes any legal
responsibility for the accuracy, completeness, or usefulness of any information,
apparatus, product, or process disclosed, or represents that its use would not
infringe privately owned rights. Reference herein to any specific commercial
product, process, or service by its trade name, trademark, manufacturer, or
otherwise, does not necessarily constitute or imply its endorsement,
recommendation, or favoring by the United States Government or any agency
thereof, or the Regents of the University of California. The views and opinions of
authors expressed herein do not necessarily state or reflect those of the United
States Government or any agency thereof or the Regents of the University of
California.

Work at Stonybrook University was supported by the Department of Energy's Office
of Science under contract number DE-SC0016605.

These simulations were performed using resources of the National Energy Research
Scientific Computing Center, a DOE Office of Science User Facility supported by the
Office of Science of the U.S. Department of Energy, also under Contract No. DE-AC02-
05CH11231.

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

**Appendix**
Figures A1 and A2 show the differences between HAPPI scenarios for the tropical
storm track and cyclogenesis densities. The top panels show the differences for each
warmer stabilized scenario minus the Historical simulation individually. The lower
left panels show the difference between the 2.0$^o$C and 1.5$^o$C stabilized scenarios. The



lower right panels show the difference between the average of the two stabilized
scenarios minus the Historical simulation.

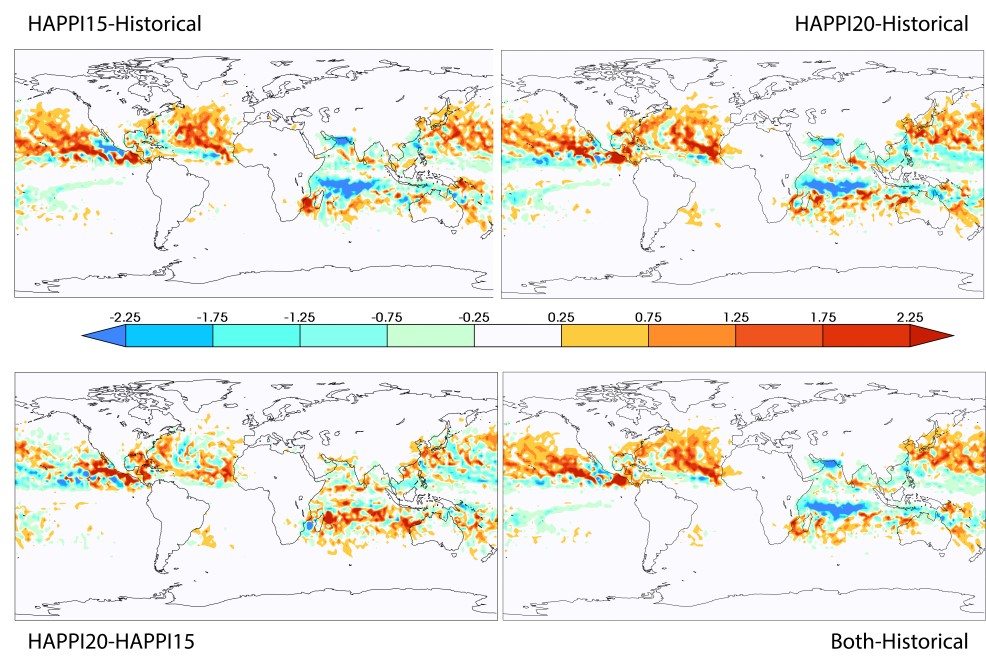

Figure A1. Percent difference of normalized tropical cyclone track density for the
HAPPI simulations.

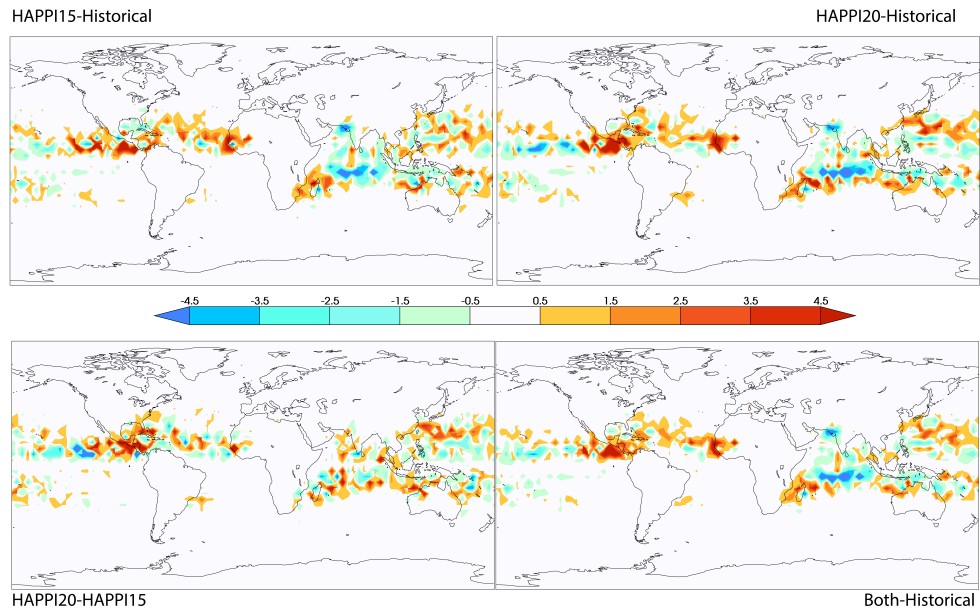



1    Figure A2. Percent difference of normalized tropical cyclogenesis density for the
2    HAPPI simulations.
