# Peer review of "Earth Syst. Dynam. Discuss., https://doi.org/10.5194/esd-2017-101"

_Earth System Dynamics, 2017_

## Referee Comment (RC1) · Anonymous Referee #1 · 19 Dec 2017

This is a nice paper that reports the results of state-of-the-art ensemble simulations of tropical cyclone formation using a high resolution climate model. The results are good although they are more along the lines of confirming what we already suspected rather than completely new results. I have only minor comments.

P.2, Line 11: "are" should be "is"

P.2., Lines 14-17: This description of the methodology is not as clear as it could be. It only became clear to me what these lines meant when I read P.3, lines 34-35. These

[Figure]

lines should be rephrased accordingly.

Figure 2: these aerosol effects appear quite large compared with the CO2 forcing and indeed the authors say this on P.13. While the HAPPI experiments are no doubt more realistic for having included the possible effects of aerosols, do the authors have any plans to assess the impact of aerosols using new simulations?

P.9, Line 10, and P.14, Line 13: It is possible to test whether these poleward shifts are statistically significant or not use a K-S test or similar.

P.9, Line 17-18. The authors are probably correct that there may be effects on extratropical transition, but a reference would assist in making their point.

---

## Referee Comment (RC2) · Anonymous Referee #2 · 19 Dec 2017

General comments: This paper describes changes in tropical cyclone (TC) activities simulated by the Community Atmospheric Model under the HAPPI protocols. The HAPPI protocol is a relatively smaller change in global mean temperature at 1.5 and 2.0 stabilizing levels compared to those mostly used for time slice experiments under global warming condition. One might have expected that the simulated signals of TCs would be smaller or not be detected. However, in this paper, the simulation shows robust changes in TC activities similar to those obtained the existing literatures. In particular, the reduction of TC number, particularly that of Category 0, over the global

domain is very robust and consistent. Projections of such extreme events as tropical cyclones under the condition of the HAPPI protocols are informative and useful for the society. This paper should be published soon with some minor revision suggested below.

The projection of the characteristics of tropical cyclones should also be tabulated. The data will be compiled later for comparison with other model results. Please refer to IPCC AR5 (Chapter 14, Supplementary Material).

Specific comments: p. 10, Fig. 5: The curve for HAPPI 1.5 is not visible. Please clarify the figure.

p. 11, Fig. 6: This figure should be compared to observation. Add bars of the observational number of ACE. Why ACE in the South Indian Ocean is so high? Please add explanation.

p. 12: "Chavas et al. (2016)" (L8) is not included in the reference list, so that the definition of "Chavas radii" (L9) is not clear. We cannot understand the following sentences in the paragraph and Fig. 8. Please explain why the radii is larger for the weaker tropical cyclones.

p. 14, L16-18, "... are not significantly altered in warmer climates, most notably the robust relationship between maximum wind speeds and central pressure minima": The conclusion of the subtle behaviors is specific to this model. In particular, pressure-wind relation likely depends on the model resolution. Such remarks should be added.
* * *

---

## Author Comment (AC1) · 17 Jan 2018

Note responses are in red.

This is a nice paper that reports the results of state-of-the-art ensemble simulations of tropical cyclone formation using a high resolution climate model. The results are good although they are more along the lines of confirming what we already suspected rather than completely new results. I have only minor comments. P.2, Line 11: "are" should be "is" Done.

[Figure]

P.2., Lines 14-17: This description of the methodology is not as clear as it could be. It only became clear to me what these lines meant when I read P.3, lines 34-35. These lines should be rephrased accordingly. We changed the wording to "A stabilized anthropogenic climate change to these surface forcing functions is constant in time. By adding such a change to the observations, observed interannual variations are preserved. "

Figure 2: these aerosol effects appear quite large compared with the CO2 forcing and indeed the authors say this on P.13. While the HAPPI experiments are no doubt more realistic for having included the possible effects of aerosols, do the authors have any plans to assess the impact of aerosols using new simulations? It is on the list of things to do, behind simulations at higher levels of warming. This is a topic that should be discussed by the HAPPI principals, in order to develop an experimental protocol as this forcing is also important for other extreme events such as heat waves.

P.9, Line 10, and P.14, Line 13: It is possible to test whether these poleward shifts are statistically significant or not use a K-S test or similar. We tested the statistical significance of the poleward shift of the normalized track density by calculating its zonal mean for each individual year of each realization separately. Because there are so many years, the standard errors in these figures are quite small. The figure below shows the normalized track density with the lines widened to reflect plus and minus 1 standard error. In the region of interest, it is clear that the poleward shifts are highly statistically significant. In this calculation, we grouped the HAPPI1.5 and HAPPI2.0 simulations together but this does not affect our conclusion. We added this sentence "The statistical significance of the larger differences in normalized track density between the historical and warmer stabilized scenarios is very high as assessed by comparison of the standard errors."

P.9, Line 17-18. The authors are probably correct that there may be effects on extra-tropical transition, but a reference would assist in making their point We added Liu et al. 2017; Zarzycki et al. 2017

[Figure]

[Figure]

**Fig. 1.** Zonal track density with line width reflecting uncertainty expressed as standard error.

---

## Author Comment (AC2) · 17 Jan 2018

Replies are in red

General comments: This paper describes changes in tropical cyclone (TC) activities simulated by the Community Atmospheric Model under the HAPPI protocols. The HAPPI protocol is a relatively smaller change in global mean temperature at 1.5 and 2.0 stabilizing levels compared to those mostly used for time slice experiments under global warming condition. One might have expected that the simulated signals of TCs

would be smaller or not be detected. However, in this paper, the simulation shows robust changes in TC activities similar to those obtained the existing literatures. In particular, the reduction of TC number, particularly that of Category 0, over the global domain is very robust and consistent. Projections of such extreme events as tropical cyclones under the condition of the HAPPI protocols are informative and useful for the society. This paper should be published soon with some minor revision suggested below.

The projection of the characteristics of tropical cyclones should also be tabulated. The data will be compiled later for comparison with other model results. Please refer to IPCC AR5 (Chapter 14, Supplementary Material). We added tables to this effect to the Appendix.

Specific comments: p. 10, Fig. 5: The curve for HAPPI 1.5 is not visible. Please clarify the figure. Colors have been changed. The gray is now darker.

p. 11, Fig. 6: This figure should be compared to observation. Add bars of the observational number of ACE. Why ACE in the South Indian Ocean is so high? Please add explanation. We have added observational estimates of ACE according to Table 2 of Maue (2011). He did not calculate SIO ACE but did calculate southern Hemisphere. In the revised figure, the model and the observational estimate are quite consistent for the SH. However, we note that in the SH, TCs are probably not as well characterized as in the NH. Revised text added as follows:"Comparison with an observational estimate (Maue 2011) suggests that the model is overactive by this measure of tropical cyclone activity although differences in the methods that tracks and wind speeds are calculated could explain some of the biases shown in figure 6. "

Also note, there was a factor of 2 error in the original figure due to a miscommunication between the software engineer and the lead author. This has been corrected.

p. 12: "Chavas et al. (2016)" (L8) is not included in the reference list, so that the definition of "Chavas radii" (L9) is not clear. We cannot understand the following sentences

in the paragraph and Fig. 8. Please explain why the radii is larger for the weaker tropical cyclones. We have added the missing citation and reworked this paragraph to replace "storm size" with "Chavas radii". The intent is to measure structural changes in tropical cyclones and we agree that having multiple "storm sizes" for a single storm is confusing. By referring this structural quantity to the Chavas radii, we hope it is clearer. This is a topic that we aim to provide a specialized paper on later this year.

p. 14, L16-18, ": : : are not significantly altered in warmer climates, most notably the robust relationship between maximum wind speeds and central pressure minima": The conclusion of the subtle behaviors is specific to this model. In particular, pressure-wind relation likely depends on the model resolution. Such remarks should be added. We expand the discussion on this point with the following sentences: "The peak wind speeds and central pressure minima relationship is controlled by the mechanical constraints of gradient wind balance, storm size and Coriolis force (Chavas 2017 and Chavas, private communication). The small poleward shift in the track density (figure 5) and subtle structural changes in wind speed radii discussed below are not large enough to change this relationship. Warmer temperatures do change the distribution of peak wind speeds and central pressure minima (figure 3) but does not appear to substantially change how they co-vary. We do note however that model resolution and structure may influence the simulation of this relationship..."

———————————————————